# Longer internal exons tend to have more tandem repeats and more frequently experience insertions and deletions

Keiichi Homma[1], Hiroto Anbo[1], Motonori Ota[2], Satoshi Fukuchi[1]

**Insertions and deletions (indels) are known to preferentially encode intrinsically disordered regions (IDRs), regions that by themselves do not form unique three-dimensional structures. As we previously showed that long internal exons tend to encode IDRs, we decided to analyze how indels alter internal exons and affect IDRs of the encoded proteins. Here, we analyzed eight eukaryotes to select indels commonly observed in all variants ("fixed" indels). The fixed indels in internal exons mostly encode IDRs. Residue-wise ~50% of the indels are attributable to alterations in tandem repeats. Deletion is generally more prevalent in long internal exons and in most species the same trend is detected in insertion. Tandem repeats occur preferentially in long internal exons, indicating that their alterations partly account for the high frequency of indels in long internal exons. Also, since tandem repeats mostly encode IDRs, this finding partially explains the high incidence of IDRs in long internal exons. We propose that long internal exons had been produced in early eukaryotes mainly by repeat expansion that added IDRs to the encoded proteins.**

## Introduction

Only a small fraction of prokaryotic proteins (residue-wise 2.0% of archean and 4.2% of eubacterial proteins) consist of intrinsically disordered regions (IDRs) that by themselves do not form unique three-dimensional structures, whereas 33.0% of residues in eukaryotic proteins are comprised of IDRs (Ward et al, 2004). IDRs in eukaryotic proteins participate in binding to other molecules and are thereby involved in important functions such as gene transcription and signaling (Anbo et al, 2019). Two mechanisms of de novo IDR generation are expansion of repetitive DNA sequences (Tompa, 2003) and exonization of introns (Kondrashov & Koonin, 2003; Sorek, 2007; Marquez et al, 2015), but to our knowledge their proportional shares have not been quantified. It is of interest to elucidate how IDRs had been acquired in the evolutionary process

from the first eukaryotic common ancestor (FECA) to the last eukaryotic common ancestor (LECA).

We previously reported the general eukaryotic tendency of long internal exons to encode IDRs and proposed that long internal exons in the LECA had resulted from short exons by addition of IDR-encoding nucleotides (Fukuchi et al, 2023). As indels play important roles in exon length alterations and mostly encode IDRs (Light et al, 2013a, 2013b; Khan et al, 2015), indel analyses may reveal mechanisms of exon length alterations and thereby explain how the preferential encoding of IDRs by long internal exons came about.

Indels in human genomes cause as much variation as small nucleotide polymorphisms (Mullaney et al, 2010) and one way to identify them in coding sequences is to find indels in genomes and select those in coding sequences (Anzai et al, 2003; Mills et al, 2006; Wetterbom et al, 2006; Fan et al, 2007; Khan et al, 2015; Lin et al, 2017). As coding sequences vary from one alternative splicing variant to another, however, different sets of coding sequences result in different indel identifications. Another method to select indels in proteins is to compare the amino acids sequences of orthologous proteins either by sequence alignments (Benner et al, 1993; Light et al, 2013a, 2013b) or by structural alignments (Pascarella & Argos, 1992; Qian & Goldstein, 2001). Since indels thereby identified are dependent on the selection of orthologous proteins, splicing variants need to be considered, and structural alignments cannot be made to stretches containing IDRs, which is a relevant issue because, as stated above, indels frequently encode IDRs (Light et al, 2013a, 2013b; Khan et al, 2015). As the increasing availability of variant sequences makes it possible to identify indels shared by all variants, we conducted sequence alignments of all available variants, selected commonly observed indels, and called them "fixed" indels.

Fixed indels can result from constitutive changes in splicing and genomic mutations. Changes in splicing that give rise to insertions can be regarded as intron to exon conversions (exonizations), whereas those resulting in deletions can be considered as exon to intron conversions (intronizations). Indels by genomic mutations have three plausible causes: DNA slippage, DNA damage followed by imperfect repair mostly by means of homologous repair pathways (Redelings et al, 2024), and transposable elements,

[1]Program for Information Systems, Division of Informatics, Bioengineering and Bioscience, Maebashi Institute of Technology, Maebashi, Japan  [2]Graduate School of Informatics, Nagoya University, Nagoya, Japan

Correspondence: khomma@maebashi-it.ac.jp

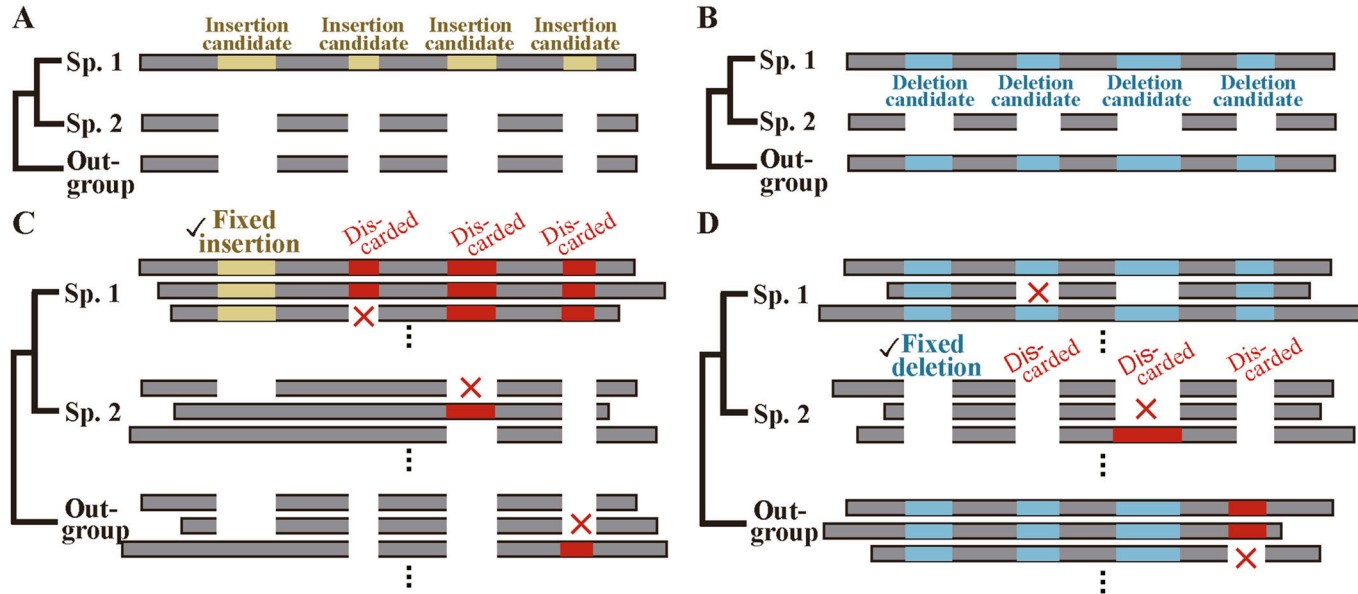

**Figure 1. Selection of fixed indels.**
**(A, C)** The criteria for selecting fixed insertions. The rectangles represent sections of coding sequences and are arranged so that aligned segments match vertically. **(B, D)** The criteria for selecting fixed deletions. They are shown as in (A, C).

which constitute ~4% of human protein-coding genes (Nekrutenko & Li, 2001). DNA slippage frequently occurs in short tandem repeats that in most cases encode IDRs (Levinson & Gutman, 1987; Simon & Hancock, 2009). Proteins in all kingdoms of life have tandem repeats, but prokaryotic proteins tend to have tandem repeats less often than eukaryotic proteins do (Delucchi et al, 2020).

Since splicing of exons longer than 300 nt is inhibited unless flanked by short introns (Robberson et al, 1990; Chen & Chasin, 1994; Sterner et al, 1996), many exons that became excessively long may not be efficiently spliced. On the other hand, analyses of RNA-Seq data demonstrated that introns shorter than 70 nt are not completely spliced out (Abebrese et al, 2017). Exon definition was proposed to explain the splicing mechanism in vertebrates in which short exons separated by long (>250 bp) introns predominate (Robberson et al, 1990; Berget, 1995; De Conti et al, 2013), whereas intron definition was postulated to account for the splicing of long exons with short introns prevalent in lower eukaryotes (Lang & Spritz, 1983; Berget, 1995; De Conti et al, 2013). Recent research disclosed that the same spliceosome assembled on introns provides the mechanisms for both intron and exon definition in *Saccharomyces cerevisiae* (Li et al, 2019).

Here, we identified fixed indels that are uniformly present in all variants and attempted to determine their generation mechanisms and found a considerable number of indels generated by alteration in the number of tandem repeats, which frequently encode IDRs. Our analyses also revealed that longer internal exons tend to experience more indels and generally have a higher prevalence of tandem repeats, suggesting that long internal exons were generated chiefly by repeat expansions. We propose that the expansion of tandem repeats encoding IDRs was a crucial evolutionary mechanism by which the LECA had arisen from the FECA.

## Results

### Selection of fixed indels in internal exons

Exploiting a wealth of sequences in the Ensembl database (Martin et al, 2023), we first aligned all variants of orthologous genes in closely related sp. 1 and sp. 2 and selected sections missing in sp. 2 as indel candidates (Fig 1A and B). We then carried out BLASTN alignments (Altschul et al, 1990) of all variants of orthologous genes in spp. 1 and 2 and an outgroup to classify them into insertion and deletion candidates; those present in an outgroup variant were considered as insertion candidates because the segments were probably absent before the bifurcation of sp. 1 and sp. 2 (Fig 1C), whereas those missing in an outgroup variant were regarded as deletion candidates as the segments were likely to be present in the immediate ancestor of sp. 1 and sp. 2 (Fig 1D). To filter out candidates that correspond to splicing variants, we applied uniformity tests, i.e., we selected the insertion candidates whose segments are present in all sp. 1 variants and are unexceptionally absent in sp. 2 and outgroup variants and regarded them as fixed insertions (Fig 1C); we likewise chose the deletion candidates whose segments are invariably absent in sp. 2, but omnipresent in both sp. 1 and outgroup variants and considered them as fixed deletions (Fig 1D). Finally, only those in internal exons are selected and are called fixed indels. We present the actual steps followed to select fixed indels in internal exons in Fig S1 and the number of cases in Table S1 and Fig S2A–H (for explanations of step 5, see below).

Insertions are probably generated either by intron to exon conversion (exonization) (Fig 2A–H) or by genomic modifications, the latter of which may be caused by expansion of tandem repeats (Fig 2I), homologous recombination (Fig 2J), or insertion of

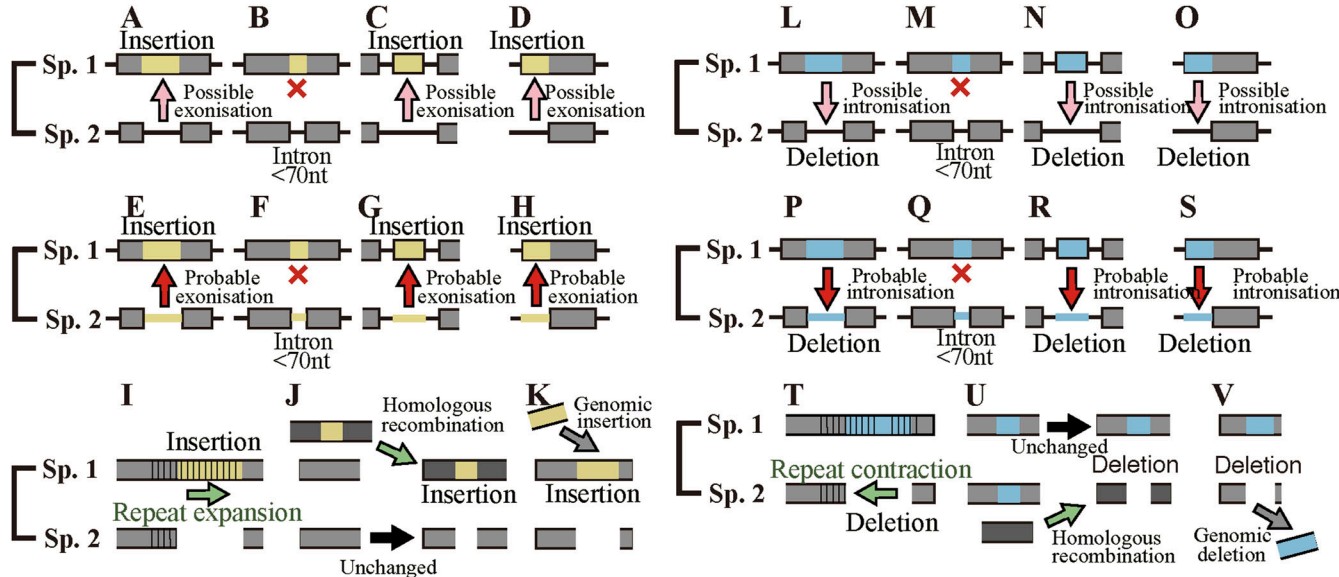

**Figure 2. Generation mechanisms of fixed indels.**
**(A, B, C, D, E, F, G, H, I, J, K)** Presumed generating mechanisms of fixed insertions. Gray rectangles and horizontal bars stand for exons and introns, respectively, whereas parallel vertical bars signify tandem repeats. **(L, M, N, O, P, Q, R, S, T, U, V)** Possible generating mechanisms of fixed deletions. They are presented as in (A, B, C, D, E, F, G, H, I, J, K).

exogenous segments (Fig 2K). Inserted segments that are presumably generated by exonization but correspond to introns with lengths less than 70 nt (Fig 2B and F) were discarded (step 5 in Fig S1) because such introns may be imperfectly removed (Abebrese et al, 2017), making them unqualified as "fixed" insertions. Similarly, the three generating mechanisms of deletions are exon to intron conversion (intronization) (Fig 2L–S), contraction of tandem repeats (Fig 2T), homologous recombination (Fig 2U), and genomic deletion (Fig 2V). Deleted segments that corresponded to short (<70 nt) introns (Fig 2M and Q) were removed for the aforementioned reason (in the following, we additionally carried out analyses of the permissive sets of fixed indels to demonstrate that the removal does not affect conclusions). Insertions with corresponding introns in sp. 2 possessing no significant homology were considered as possibly generated by exonization (Fig 2A, C, and D), whereas those alignable to intron segments in sp. 2 were classified as probably generated by exonization (Fig 2E, G, and H). As introns generally have a higher mutation rate than exons do (Li, 1997), exons created by exonization may no longer possess detectable sequence homology with the introns. Likewise, we consider deletions as possibly generated by intronization if the corresponding intron segment in sp. 2 has no discernable similarity to the exons in sp. 1 (Fig 2L, N, and O), whereas sorting those as probably generated by intronization if the intron segments were alignable to the exons (Fig 2P, R, and S). We distinguish fixed insertions of entire exon(s) (Fig 2C and G) from those within exons (Fig 2A and C) and regard fixed deletions of entire exon(s) (Fig 2N and R) separately from those within exons (Fig 2L and P). The total number of indels within internal exons as well as their breakdown by indel location within exons are shown in Table S2, Fig S3A–J, and Table S3, the last of which lists numbers without removing those corresponding to short (<70 nt) introns (designated the "permissive" sets of fixed indels). Fixed indels of entire exon(s) are

rare and almost all the fixed indels in internal exons are in the middle of rather than at 5′ and 3′ ends of exons.

Six actual examples of fixed indels are provided (Fig 3A–F) and two examples of indels that were removed due to their correspondence to excessively short (<70 nt) introns are shown (Fig S4A and B). We selected these solely based on easy visualization of indels. The human protein in Fig 3A is a protein enabled homolog involved in a range of processes dependent on cytoskeleton remodeling and cell polarity, the *Drosophila melanogaster* protein in Fig 3B is E2F transcription factor, isoform D, whereas the rat protein presented as Fig 3C is nuclear factor erythroid 2-related factor 2, which is a transcription factor that plays a key role in the response to oxidative stress. The chimpanzee protein depicted in Fig 3D is a trinucleotide repeat-containing protein that plays a role in RNA-mediated gene silencing, the human protein shown in Fig 3E is GRIP and coiled-coil domain-containing protein isoform 2, whereas the human protein in Fig 3F is SMG1 phosphatidylinositol 3-kinase-related kinase partaking in both mRNA surveillance and genotoxic stress response pathways.

### Generation mechanisms of fixed indels

The residue-wise distributions of generation mechanisms of fixed indels are graphically presented (Fig 4) with the indels whose generating mechanisms remain unidentified labeled "unknown." No indel cases involving transposable elements were detected even if the expectation value of the BLASTN search was relaxed to 10 and no insertion cases whose generation mechanisms are unknown were long enough to be assessed for involvement of other exogenous elements. The corresponding figure of the permissive sets of fixed indels is presented as Fig S5. The numbers of fixed indel cases and those of residues in insertion and deletion

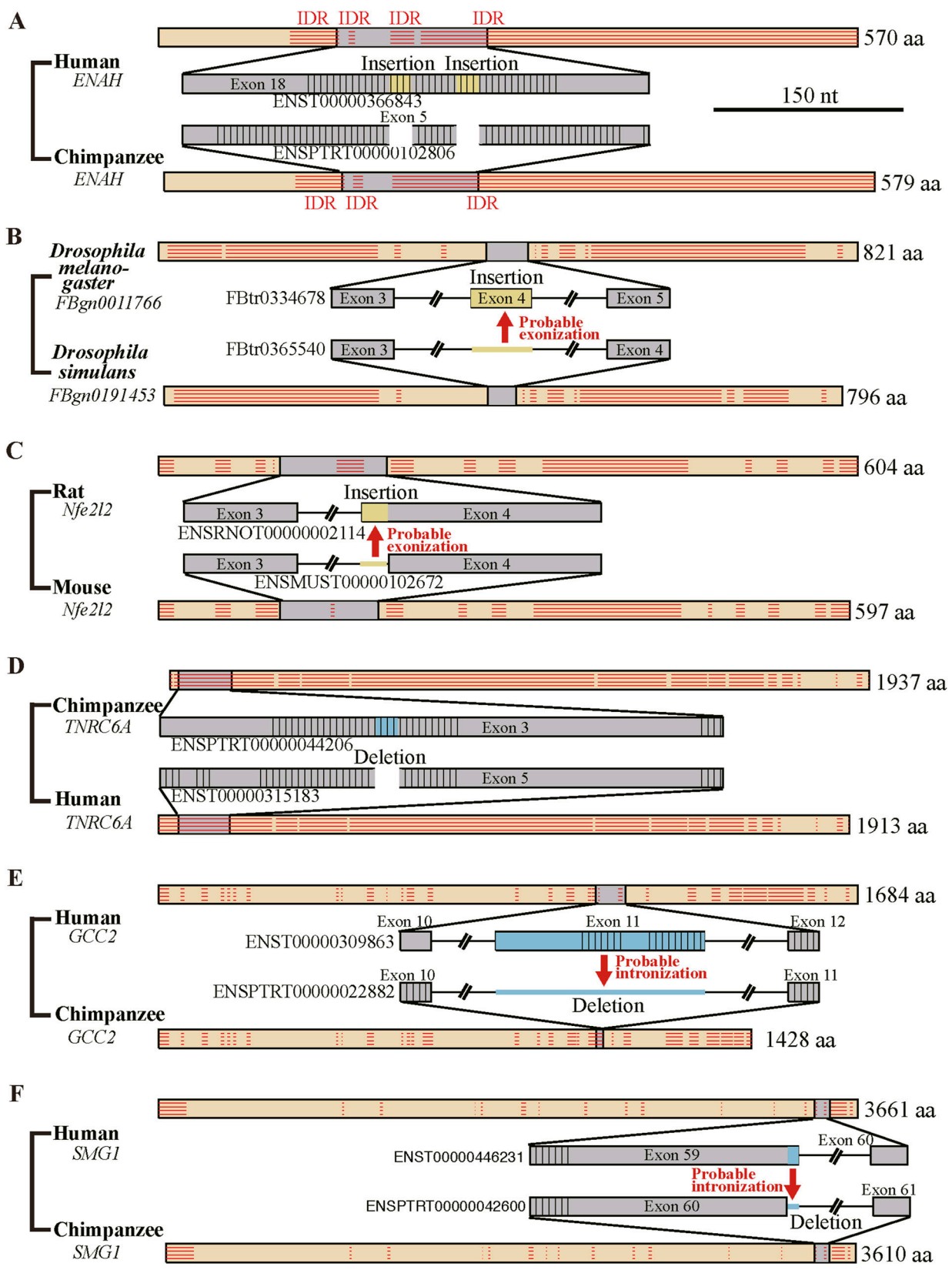

are tabulated (Table S4), whereas the corresponding numbers of the permissive sets of indels are presented as Table S5. Although interspecies variations are considerable, on average 50.9% of inserted and 49.8% of deleted residues were attributable to alterations in tandem repeats and the generation mechanisms of a substantial number of fixed indels remain unidentified (see the Discussion section). Fixed indels generated by intronization and exonization exist but have small shares except in human insertions, chimpanzee deletions, and rat, *O. latipes*, and *O. melastigma* indels whereas those attributable to homologous recombination are universally rare.

## Long internal exons tend to have a high incidence of indels

As we previously found that long internal exons tend to encode IDRs (Fukuchi et al, 2023), we searched for generation mechanisms of long internal exons and thus determined the dependence of indels on exon length. We first calculated the abundance of internal exons in 60 nt length bins (Fig S6A–I) and found that, whereas long internal exons are rare in the four mammals and the two fishes, the two *Drosophila* species showed length distributions shifted rightward, confirming a previous report (Hawkins, 1988). We then examined the frequencies of indels in each length bin of internal exons, using the length of the corresponding sp. 1 exons for the analyses of insertion in sp. 1 and deletion in sp. 2. The length of the internal exon with an insertion used was that of the exon minus the insertion length because that is the presumed exon length before insertion. In the example presented as Fig 3C in which a 21 nt insertion exists in rat exon 4 of 189 nt, we used 168 nt as the length of the primordial exon in which the insertion occurred (although 168 nt happens to be the length of the corresponding sp. 2 exon in this case, the lengths sometimes disagree as some sp. 2 exons have been altered). The indels of entire exon(s) (Fig 2C, G, N, and R), which constitute small fractions (Fig S3, Tables S2 and S3), were not included in this and subsequent analyses because our aim was to analyze incremental length changes in internal exons. Interestingly, we found higher frequencies of deletions in long internal exons without exception and the insertion frequencies of five out of eight species were positively correlated with internal exon length (Figs 5B–I and S7B–I). The three species that did not show a large positive correlation in insertion frequency were chimpanzee, *Oryzias laptipes*, *Oryzias melatigma*, and *O simulans*. As the smallness of samples may account for the lack of correlation, we calculated the indel frequencies in three length ranges of internal exons, namely S (1–180 nt), M (181–360 nt), and L (361–540 nt) and compared them. All but the insertion frequencies of *O. laptipes* and *O. melatigma* showed a statistically significant increase in L compared with M (Figs S8B–I and S9B–I). Thus, indel frequencies generally increase with exon length, except for the two fish species in which insertion frequencies do not appreciably vary with exon length. In addition, overall deletion frequency is higher than insertion frequency in all the species examined (statistically significant at $P$ value $< 10^{-3}$, chi-square test). That deletion generally exceeds insertion agrees with previous reports (de Jong & Rydén, 1981; Redelings et al, 2024).

We also calculated the average indel frequency of the eight species in each internal exon length bin. The combined dependence of indel frequency on internal exon length (Figs 5A and S7A) shows that positive correlations exist although the correlation of insertion is weaker than that of deletion. The analysis in the three exon length ranges supports the observation (Figs S8A and S9A). Additionally, deletion frequency is on average higher than insertion frequency with the difference more pronounced in long internal exons: the overall ratio of deletion frequency to insertion frequency is 2.72 in range S (1–180 nt), increases to 4.07 in range M (181–360 nt), but remains at 4.07 in L (361–540 nt).

## Most fixed indels encode intrinsically disordered regions, conserve reading frames, and are short

As indels reportedly occur preferentially in IDRs (Light et al, 2013a, 2013b; Khan et al, 2015), we calculated the fractions of IDRs in the fixed indels. For deleted segments in sp. 2, we used the corresponding segments in sp. 1 (Fig 1D) as a proxy. In agreement with the literature, most indels encode IDRs and the fractions are all much higher than those of all coding exons (Figs 6A, S10A and B, and S11A and B). We also examined indel lengths and verified reports (Mills et al, 2006; Wetterbom et al, 2006; Khan et al, 2015) that those of nearly all indels in coding sequences are multiples of 3 nt (Figs 6B, S12, and S13). Frame-preserving insertions except for those containing stop codons merely add residues to the encoding proteins, whereas frame-preserving deletions excluding those containing start or stop codons simply remove residues. Since most indels encode IDRs, alterations in the number of amino acids caused by fixed indels are likely to be tolerated. The length distributions (Figs 6C, S14A–I, and S15A–I) demonstrate that shorter indels predominate, in agreement with reported findings (Benner et al, 1993; Qian & Goldstein, 2001; Khan et al, 2015) and almost all fixed indels are shorter than 19 nt.

## Amino acid compositions of indels resemble those of IDRs

As most indels encode IDRs and IDRs have a characteristic amino acid composition (Dunker et al, 2008), we calculated the amino acid compositions of indels (Figs 6D and S16). Indels have amino acid compositions similar to those of IDRs; in indels order-promoting residues are depleted, whereas disorder-promoting residues appear more frequently than the average. For the data presented, the correlation coefficient between log fold changes of insertion and those of IDR is 0.797 (statistically significant at $P$ value $< 10^{-4}$),

**Figure 3. Actual examples of fixed indels.**
**(A, B, C)** Examples of fixed insertions. They are depicted as in Fig 2 in the scale indicated with tandem repeats represented by parallel vertical bars. The gene IDs are shown in italics, whereas the encoded proteins are drawn as orange rectangles in arbitrary scales with horizontal parallel bars representing intrinsically disordered regions and gray rectangles corresponding to the depicted genomic sections. **(D, E, F)** Examples of fixed deletions. They are drawn as in (A, B, C).

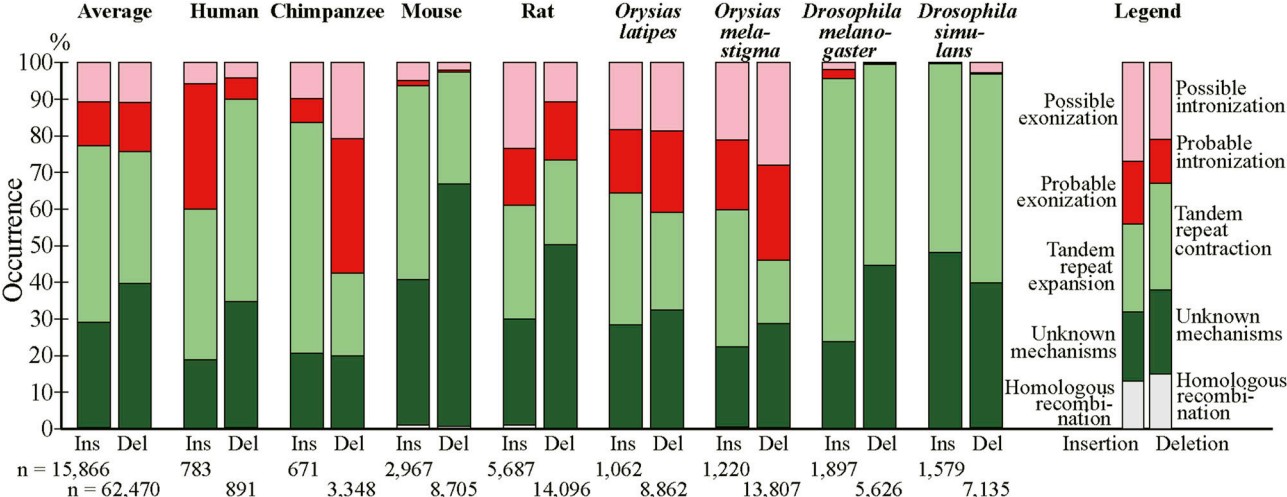

**Figure 4. Generating mechanisms of fixed indels.**
The numbers of nucleotides in indels are shown at the bottom.
Source data are available for this figure.

whereas the correlation coefficient between log fold changes of deletion and those of IDR is 0.876 (statistically significant at $P$ value $< 10^{-5}$).

## Long internal exons tend to have tandem repeats

What accounts for the higher incidences of indels in longer internal exons? We found no grounds to suppose that intronization, exonization, and homologous recombination occur more frequently in longer internal exons. Since tandem repeats account for a large fraction of indels (Fig 4) and longer internal exons generally experience more indels (Fig 5A–I), we thought it possible that tandem repeats are more prevalent in long internal exons. We thus calculated the fraction of each internal exon occupied by tandem repeats and investigated its dependence on internal exon length. The results (Fig 7A–I) verified the conjecture; the longer internal exons are, the higher the repeat frequency with all the correlation coefficients statistically significant at $P$ value $< 10^{-2}$ and the conclusion remains unchanged in the longer length range (1–840 nt instead of 1–540 nt) (Fig S17A–I).

To examine how much of the higher prevalence of IDRs in longer internal exons is explained by tandem repeats, we calculated the fractions of IDRs, tandem repeats within IDRs, and others within IDRs (i.e., IDRs that are not in tandem repeat segments) in each length bin in the eight species in the 1–540 nt (Fig 7A–I) and 1–840 nt length ranges (Fig S17A–I). Note that the previously reported dependence of the fraction of IDRs on internal exon length (Fukuchi et al, 2023) was more pronounced in the longer exon length range. Since the IDRs are divided into tandem repeats (within IDRs) and others, the sum of the fractions of the two constituents is equal to that of IDRs and, consequently, the sum of the slopes of the regression lines of the two is identical to that of IDRs. If the positive correlation of IDRs is entirely accounted for by tandem repeats, the slope of the regression line of tandem repeats within IDRs will be the same as that of IDRs, whereas the slope of others will be zero, that is, the fraction of others will be constant irrespective of intron length. The observed slope of the

regression line of the fraction of tandem repeats in IDRs was 0.890 per cent/unit, where 1 unit equals 60 nt, and that of others in IDRs was 1.457, whereas that of the corresponding slope of the fraction of IDRs was 2.348 (Fig 7A). Thus, only 37.9% of the dependence of IDRs on internal exon length is contributed by tandem repeats. As tandem repeats constitute a mere 29.6% of IDRs on average, however, they make a disproportionate contribution to IDRs. On the other hand, in the longer range of exon length (Fig S17A), the contribution ratio of tandem repeats in IDRs was 34.0%, slightly lower than the figure calculated in the shorter exon length range (37.9%).

## Gene ontology (GO) analyses of genes with indels

What functions do genes with indels frequently have? Using SwissProt annotations, we analyzed the frequency of GO appearance in human and mouse genes with or without indels. We tabulated GO numbers that are assigned significantly more often to both human and mouse genes with indels compared with those without indels (Table S6). The genes with insertion are enriched with many functions in the nucleus such as acetyl transferase activity and RNA uridylyltransferase activity, and those with deletions also have a high prevalence of nucleus-related functions including histone kinase activity, the Las1 complex, and RNA cap trimethylguanosine synthase activity (nuclear-related GO terms are rubricated in the table).

## Involvement of indels in human diseases

Are there any indels that occur in sites related to human diseases? We consulted SwissProt annotations to examine if any of the inserted residues and the bordering residues of deletions are in known disease-related sites. We found 6 and 12 insertions and deletions, respectively, that fall on disease-related sites (Table S7). Thus, a considerable number of indels occur in sensitive locations whose mutations cause human diseases. As indels predominantly encode IDRs (Fig 6A), we determined if the disease-related indel

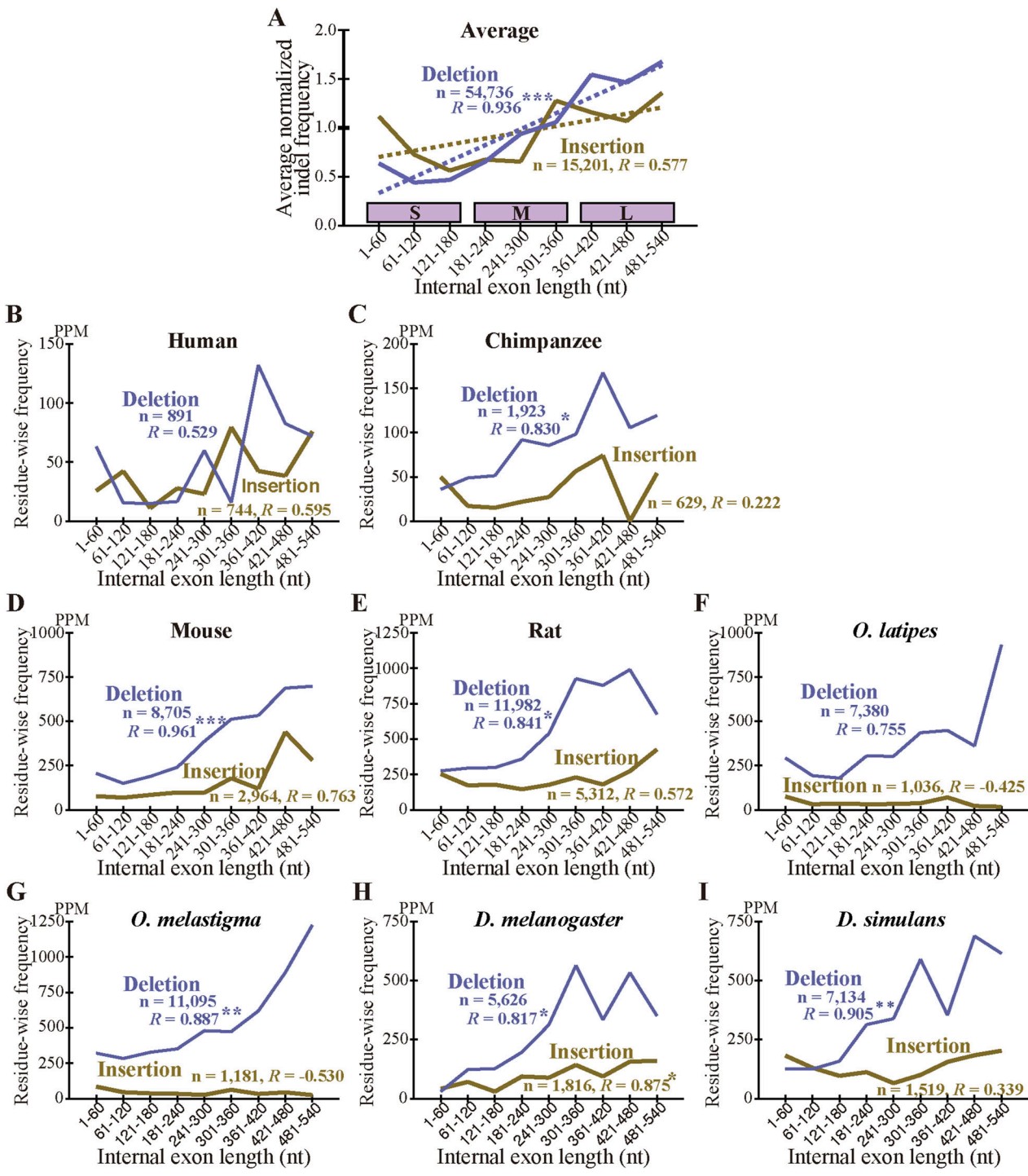

**Figure 5. Long internal exons tend to have a high incidence of indels in each species.**
**(A)** Normalized averages of indels. The insertion (left scale) and deletion (right scale) frequencies are shown with the total numbers of indel residues and the correlation coefficients. One, two, and three asterisks signify that the correlation coefficient is significantly different from zero at $P$ value $< 10^{-2}$, $10^{-3}$, and $10^{-4}$, respectively. The dotted lines are regression lines for insertion (yellow) and deletion (blue) frequencies. The length ranges (S, M, and L) used in Figs S8 and S9 are shown as the rectangles. **(B, C, D, E, F, G, H, I)** Indel frequencies in each species. **(A)** The total numbers of nucleotides in indels, the correlation coefficients, and statistical significance are displayed as in (A).
Source data are available for this figure.

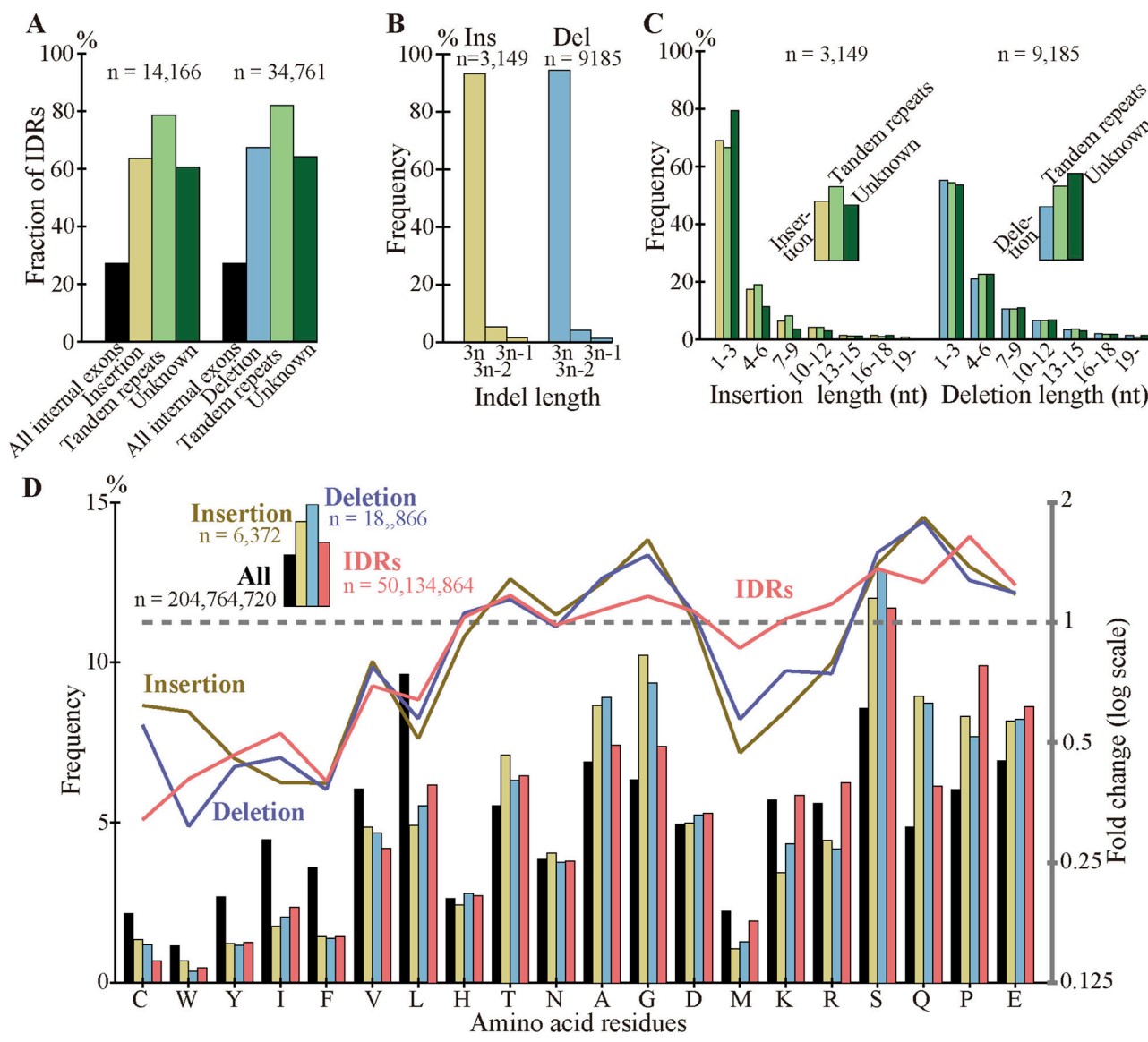

**Figure 6. Most indels in internal exons encode intrinsically disordered regions (IDRs), are multiples of 3 nt in length, and are short, and amino acid compositions of indels resemble those of IDRs.**
The bars and line graphs represent arithmetic averages of the eight species. **(A)** Fractions of IDRs. The bars represent the fractions of IDRs encoded by all internal exons, indels, tandem repeats in indels, and indels whose generation mechanisms are unknown. The total numbers of nucleotides in indels are displayed, too. **(B)** Phase distributions of indel lengths. The arithmetic averages of case-wise phase distributions of the eight species are graphed and the total case numbers are shown. **(C)** Length distributions of indels. The averages of all indels, tandem repeats in indels, and indels of unknown generation mechanisms were calculated as in (B) and presented together with the total case numbers of indels. **(D)** Average frequencies of amino acids and fold changes. The average frequency of each amino acid encoded by all internal exons ("All"), that of fixed indels, and that of IDRs are represented by rectangles (left scale). Each datum is the arithmetic average of the frequencies of the eight species. The line graphs (right scale) are the fold changes in frequency relative to that of all internal exons. Also shown are the numbers of amino acid residues encoded by all internal exons, indels, and in IDRs encoded by internal exons.
Source data are available for this figure.

segments encode IDRs and found that 12 out of the 18 segments are within IDRs (rubricated in Table S7).

## The frequency of indels correlates with that of tandem repeats

Since both the frequency of indels and that of tandem repeats mostly show a positive correlation with internal exon length, the two frequencies are likely to be correlated. The correlations between the normalized insertion/deletion frequency and the normalized repeat frequency are shown by scatter plots (Fig 8A). The correlation coefficient between insertion and repeat frequencies is 0.733 (statistically significant at $P$ value < 0.05), whereas that between deletion and repeat frequencies is 0.965 (statistically significant at $P$ value < $10^{-5}$). Similar correlations are observed in the

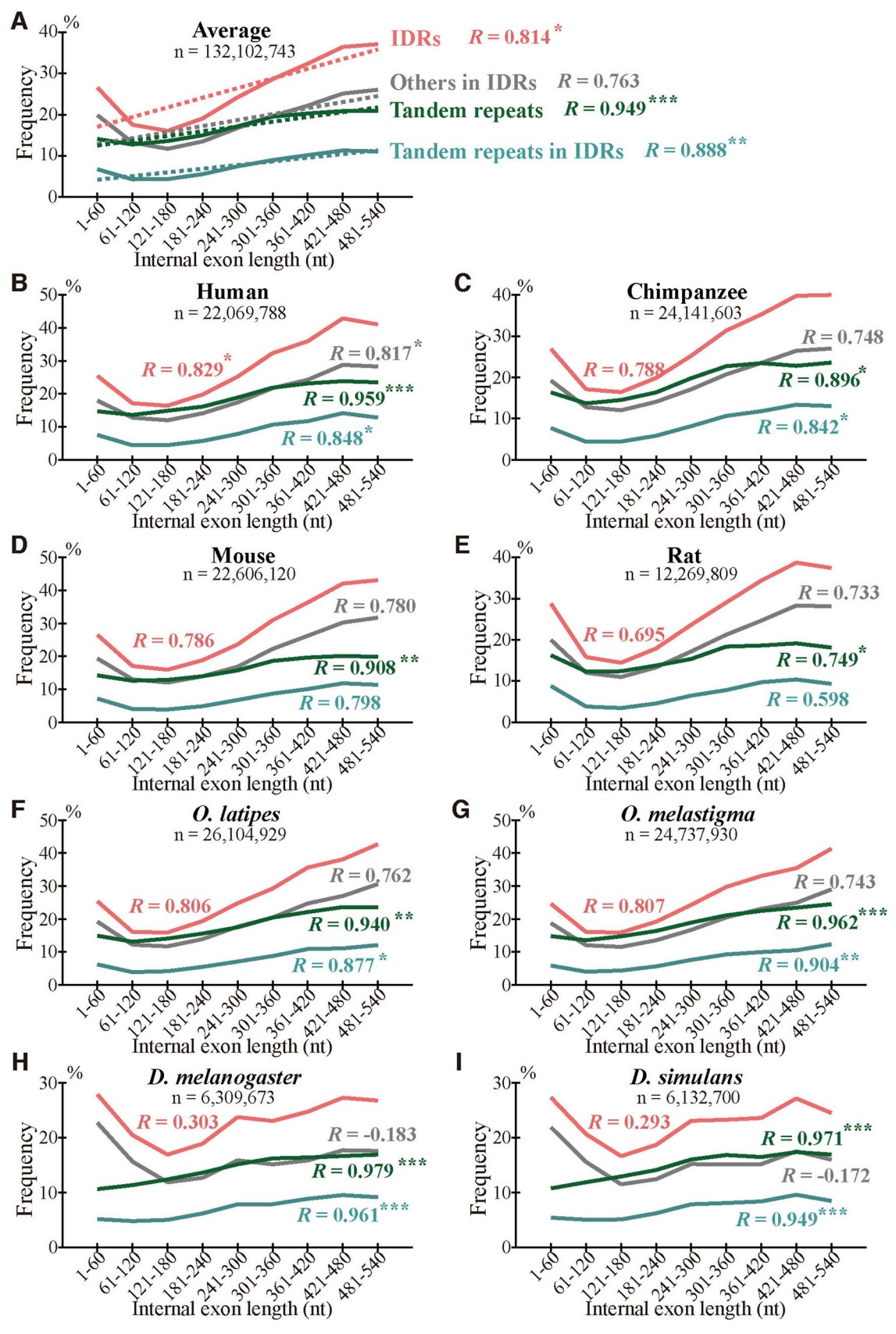

permissive sets of indels (Fig S18). In confirmation of the expectation, the indel frequencies are significantly correlated with repeat frequency, with the deletion frequency more strongly so.

# Discussion

We unexpectedly discovered that fixed indels occur preferentially in longer internal exons and large fractions of indels were generated by alterations in tandem repeats. We also found a high prevalence of tandem repeats in long internal exons, which at least partially accounts for the previously observed tendency of long internal exons to encode IDRs (Fukuchi et al, 2023) as tandem repeats mostly encode IDRs (Simon & Hancock, 2009). In view of the findings of this research, we propose that long internal exons had resulted from primordial short exons (Fig 8C) that had evolved from the FECA (Fig 8B) mainly by expansion of tandem repeats that added IDRs to the encoded proteins, which constitutes a step toward the emergence of the LECA (Fig 8D). This model explains how eukaryotic proteins acquired IDRs, especially in sections encoded by long internal exons, whereas only a small fraction of prokaryotic proteins consists of IDRs. This step also accounts for the increased frequency of tandem repeats in eukaryotic proteins compared with prokaryotic proteins. Moreover, we suggest that, after the LECA, long internal exons have been subject to frequent contractions and, to a lesser extent, expansions due to alterations in tandem repeats (Fig 8E). As nearly all fixed indels are non-frameshifting (Fig 6B), indels generated by tandem repeats mostly result in tandem repeats in proteins. Since protein tandem repeats play crucial roles in ligand binding and transcriptional regulation (Kobe & Kajava, 2001; Yagi & Nakamura, 2014), indels generated by tandem repeats may have important biological significance. Tandem repeats are rare in prokaryotes possibly because many IDRs do not play beneficial roles and thus are frequently not tolerated. The tolerability of tandem repeats in eukaryotes may be related to the splicing efficiency of long internal exons; tandem repeats that result in long internal exons are not tolerated in organisms that cannot splice long exons efficiently, and vice versa.

The two fish species are exceptional in that the insertion frequency does not show dependency on internal exon length (Figs 5F and G, S7F and G, S8F and G, and S9F and G). We note that the insertion frequencies in these species are much smaller than the deletion frequencies. The scarcity of inserted residues in the two species might account for the finding that is inconsistent with that of the remaining six species. Further research with other species is needed to clarify this issue.

The reported number, 262, of human indels in all coding exons (Mills et al, 2006) is the sum of human insertions, chimpanzee deletions (the two constitute apparent human insertions), human deletions, and chimpanzee insertions (the latter two account for apparent human deletions). The corresponding number in this investigation is thus the total of insertions and deletions in humans and chimpanzees in all exons and is 1,502 (permissive sets, step 5 in Table S2). The increase in number is attributable not only to revised sequence data we had the good fortune to use, but also to the methodology that identifies changed alternative splicing in addition to genomic alterations in constitutively spliced exons.

Although the frequencies of the detected fixed indels are small (less than 0.06%), they may well be an underestimate as our methods of selecting fixed indels probably capture only a fraction of actual fixed indels. This is both because transcripts with corresponding segments must be present in all three species in our identification method and because BLASTN alignments can be made only for highly conserved nucleotide sequences. Additionally, the outgroup requirement means that the capture rate of fixed indels varies from one species group from another and thus makes it inappropriate to compare indel frequencies of different groups; we for instance cannot meaningfully compare the indel frequencies of human and mouse. Moreover, the availability of more variant data may disqualify some fixed indels due to the presence of counterexamples in the newly identified variants. Since the methodology has no bias on exon length, however, the finding that higher indel frequency is observed in longer internal exons is unaffected by these shortcomings.

We consider the fractions of fixed indels generated by tandem repeats in exons represent an underestimate; slight sequence changes after repeat expansion and contraction make tandem repeats imperfect, making them undetectable by the repeat detection program. It is thus conceivable that indels generated by unknown mechanisms contain those attributable to tandem repeats, although they may well include some cases generated by inverted and mirror repeats (Burssed et al, 2022), which are undetectable by the program we used. In fact, the fractions of IDRs in indels generated by alterations in tandem repeats are comparable to those produced by unknown mechanisms (Fig 6A), their length distributions are nearly identical (Fig 6C), and amino acid compositions of the two groups are similar (Fig S19A and B). It is thus plausible that many indels whose generation mechanisms remain unidentified were really generated by alterations in tandem repeats. An underestimation of tandem repeats can explain why the tandem repeats explain only 38% (in the internal exon length range 1–540 nt, Fig 7A) or even less in the longer (1–840 nt, Fig S17A) exon length range of the dependence of IDRs on the length of internal exons; unidentified tandem repeats may account for some of the remaining dependence.

Notwithstanding all the emphasis on tandem repeats, we would like to reiterate that intronization and exonization also generate deletions and insertions, respectively, and they account for considerable fractions of insertions in *Homo sapiens*, deletions in chimpanzee, and indels in rat and the two fishes (Fig 4). The fact

**Figure 7. Long internal exons tend to have tandem repeats as well as intrinsically disordered regions (IDRs).**
**(A)** Arithmetic averages, regression lines, and correlation coefficients. The fractions of tandem repeats, IDRs, tandem repeats in IDRs, and others in IDRs are shown with the correlation coefficients of the fractions with exon length. Asterisks signify statistical significances as in Fig 5 legend. The dotted lines represent regression lines, whereas the sample number represents the sum of nucleotides in internal exons. **(B, C, D, E, F, G, H, I)** Frequencies and correlation coefficients in each species. The data are presented as in (A) without regression lines. The sample number below each species name is the total number of nucleotides in internal exons. Source data are available for this figure.

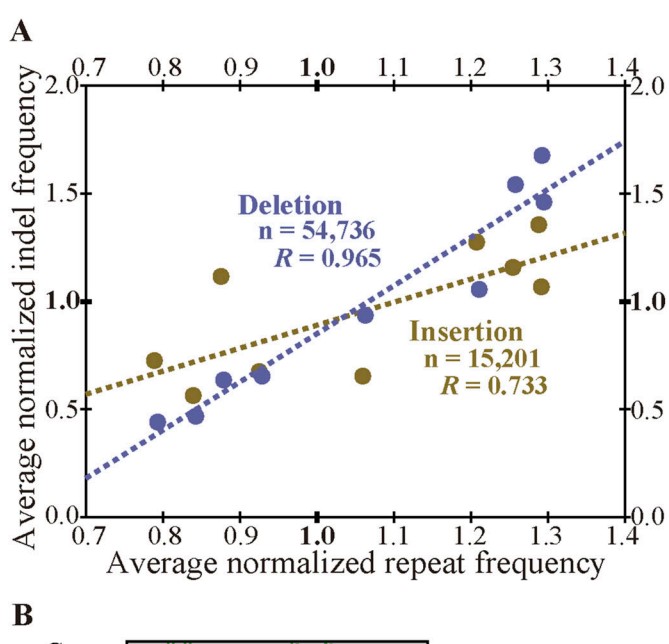

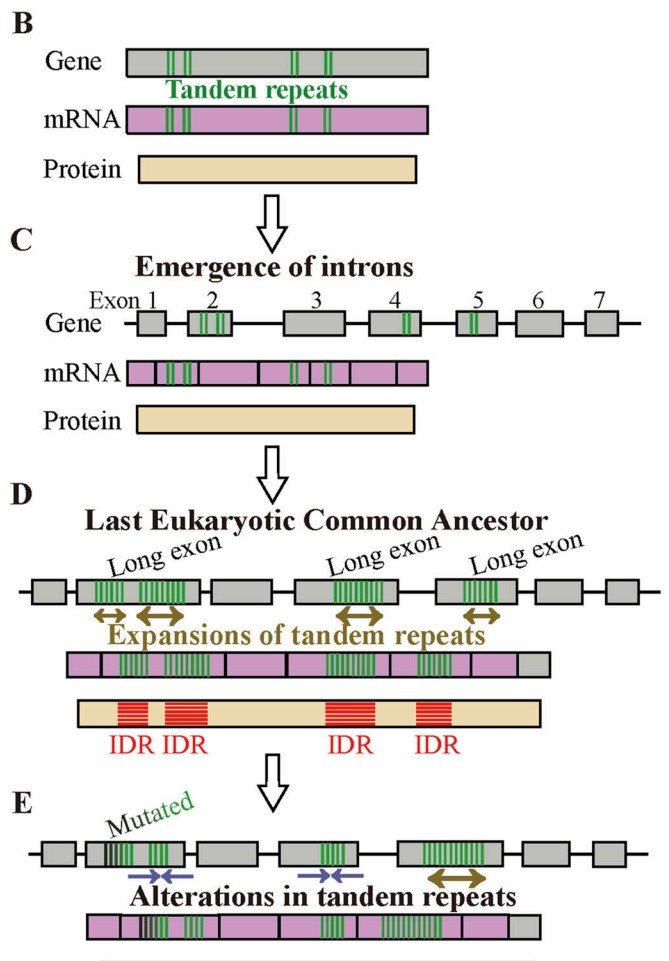

**Figure 8. Indel frequencies are correlated with tandem repeat frequency and a proposed model.**
**(A)** Correlations of repeat frequency and indel frequencies. The panel represents scatter plots of the normalized repeat frequency of each length bin and the corresponding normalized insertion (yellow dots) and deletion (blue dots)

that many indels were "probably" generated by intronization and exonization makes it quite likely that they really account for some indels. The wide variation in the fractions of intronization and exonization is largely explainable by disparity in location distribution of indels within internal exons; most insertions at 5' and 3' ends of exons and those involving entire exons were judged to be generated by exonization (87.8, 85.1, and 74.9%, respectively), and nearly all deletions at 5' and 3' ends of exons and those of entire exon(s) were attributable to intronization (93.4, 92.1, and 97.8%, respectively). By contrast, exonization and intronization were seldom assigned as the generation mechanisms of indels within exons (0.7% and 2.0%, respectively). The insertions in human, the deletions in chimpanzee, and the indels in rat and the two fishes all have high total proportions of those at the ends of exon and those involving entire exons (Fig S3C, D, F, G, and H). Consistently, these have comparatively high fractions of cases generated by intronization/exonization. Indel analyses of other species are needed to judge whether intronization and exonization generally generate less indels than tandem repeats do.

The number of insertion cases attributable to homologous recombination is much smaller than that generated by tandem repeats (Fig 4 and Table S4). This observation indicates that replication slippage rather than homologous recombination is the dominant mechanism of tandem repeat expansion.

The total absence of transposable elements in the selected indels was unexpected. Since we carried out BLASTN alignments of the entire variants with indels, transposable elements must have been detected in indels if the sequences are sufficiently similar. Possibly transposable elements that gave rise to indels had mutated considerably to escape detection by BLASTN. Since most exons containing transposable elements are alternatively spliced (Sorek et al, 2002; Lin et al, 2008), another plausible explanation is that we have filtered out those corresponding to transposable elements as we selected for "fixed" indels commonly shared by all splicing variants in our methodology.

What accounts for the general preponderance of deletion over insertion? We consider it possible that insertion is evolutionarily disfavored especially in long internal exons because splicing of many excessively long exons (>300 nt) is inhibited (Robberson et al, 1990; Chen & Chasin, 1994; Sterner et al, 1996). This interpretation dovetails with the observed tendencies that insertion frequency of internal exons initially rises but levels off at around 360 nt, whereas deletion frequency shows a monotonically increasing trend (Fig 5A). However, we excluded fixed insertions that create entire exons (Fig 2C and G) and fixed deletions that eliminate whole exons (Fig 2N and R) from our analyses of exon length dependence. Neither can our analysis capture cases of exon fusion and fission that do not affect coding sequences because our selection pipeline only detects cases whose coding sequences are affected. Although

frequencies together with the correlation coefficients. Dotted lines (yellow: insertion, blue: deletion) are regression lines. The correlation coefficients and the total numbers of nucleotides in indels are also shown. **(B, C, D)** Proposed evolutionary steps that had led to the last eukaryotic common ancestor that have long internal exons that frequently encode intrinsically disordered regions. **(E)** Ongoing process.
Source data are available for this figure.

analysis of exon evolution including these neglected cases is expected to elucidate exon evolution dynamics in its entirety, it is beyond the scope of the current study.

The gene ontology analysis revealed that as many as half of the functions significantly enriched in the genes with indels are associated with the nucleus. Since most indels encode IDRs (Fig 6A) and IDRs are especially prevalent in transcription factors and other nuclear proteins (Anbo et al, 2019), many genes with indels probably encode proteins that function in the nucleus. What accounts for the finding that as many as 18 human indels correspond to disease-related sites? Considering the enrichment of IDRs in disease-related proteins (Anbo et al, 2019) and the fact that two-thirds of the disease-related indel segments encode IDRs, we consider it likely that the high coincidence of indels with disease-related sites reflects frequent involvement of IDRs in diseases.

The existence of a positive correlation between tandem repeat frequency and internal exon length was an intriguing finding. Since four other eukaryotes, *Oryza sativa*, *Arabidopsis thaliana*, *Caenorhabditis elegans*, and *Schizosaccharomyces pombe* were shown to have a positive correlation of the fraction of IDRs and internal exon length (Fukuchi et al, 2023), we checked if repeat frequency and internal exon length are correlated in the 1–840 nt range in these model eukaryotes, too. We found the first three model eukaryotes exhibit statistically significant ($P$ value < $10^{-2}$) positive correlations, but *S. pombe* does not (Fig S20). The case of *S. pombe* is apparently inconsistent with the notion that long internal exons were produced mainly by the expansion of tandem repeats. However, as stated above, the fraction of tandem repeats is likely to be underestimated, and the underestimation can explain the lack of correlation in the yeast species.

As far as we are aware, this represents the first systematic identification of fixed indels including those generated by constitutive alterations in alternative splicing and examination of the dependence of their frequencies on internal exon length. As our observations are limited to indels in eight species only, however, the generality of our findings needs to be tested by indels in other species. Unfortunately, it is not easy to apply the current methodology to many other species because of the current selection method requires the presence of a trio of closely related species with completely sequenced genomes and an abundance of sequenced variants. For instance, the application of our methods to *C. elegans*, *Caenorhabditis briggsae* with *Caenorhabditis japonica* as the outgroup identified less than 70 cases of fixed insertions in *C. elegans*, a number judged too small for further analyses. Hopefully, widespread availability of variant and genome sequences will make extensive testing possible.

# Materials and Methods

### Data sets

For the selection of *H. sapiens* and *Pan troglodytes* (chimpanzee) indels, we chose *Gorilla gorilla* as the outgroup. As the outgroup of *Mus musculus* (mouse) and *Rattus norvegicus* (rat), we used *Peromyscus maniculatus*, whereas *Nothobranchius furzeri* (turquoise killifish) served as the outgroup of *Oryza latipes* (Japanese medaka) and *Oryza melastigma* (Indian medaka). As the outgroup of *D.*

*melanogaster* and *Drosophila simulans*, we chose *Drosophila yakuba*. We downloaded all the sequence and exon data, and the lists of orthologous genes except for that between *D. melanogaster* and *D. simulans* and the three fish species from the Ensembl database (Martin et al, 2023). Since lists of orthologous genes between the two *Drosophila* species and the fish species were unavailable from the Ensemble database, we selected pairs of orthologous genes by mutual best hit of BLASTN alignments of the longest variants. The list of transposable elements used was in release 19 of the TREP Transposable Element Platform (Wicker et al, 2022).

### Selection of fixed indels and their classification by location

The actual selection steps followed are schematically shown (Fig S1). The first two steps are the same for the selection of insertions and deletions. In step 1, we first got the DNA sequences of all transcript pairs corresponding to all the orthologous genes of sp. 1 and sp. 2. We then performed BLASTN alignments (Altschul et al, 1990) of the pairs with the default parameters except that we set the gap-opening penalty at 10 and changed the expectation value to $10^{-3}$, unless otherwise stated. We identified indel candidates in BLASTN alignments with special considerations for sections of multiple alignments; we regarded two aligned sections continuous if the last query residue of one section coincides with the first query residue of the other within 3 nt (The 3 nt allowance was made to cope with the uncertainty in alignments. The specific number was chosen by inspection of a number of alignments involving indels). We judged indel sections identical if their genomic addresses agreed within 3 nt. In the uniformity tests, we disregarded variants that do not have sections covering indels. An indel was regarded as coinciding with the 5′ or 3′ end of an exon of sp. 1 if the start or the end of the segment coincides with the start or the end of the exon within 3 nt. We judged an indel without coincidence with exon border(s) to be located inside an exon. After identifying indel candidates, we assigned the corresponding genome addresses to all the sp. 1 variants with indel candidates and made the indel candidates nonredundant based on the genome addresses. Two candidates were considered identical if either the beginning or the end genome address of one insertion candidate matches the start or the end of another within 3 nt.

In step 2, we checked all the alignments made in step 1 containing indel candidates and left only those with 0% coverage in all sp. 2 variants.

In step 3, we first collected the DNA sequences of all the variants of the sp.1 genes with indel candidates and those of the outgroup variants of the orthologous genes then ran BLASTN alignments with the same parameters as in step 1. To select insertion candidates, we then analyzed the alignments and left only the indel candidates whose coverage of the corresponding sections of the outgroup variants was 0% without exception to get insertion candidates. On the other hand, to choose deletion candidates, we selected for the candidates whose corresponding sections in the outgroup variants were universally 100%.

To select insertion candidates in step 4, we identified all the sp. 1 variants of the genes containing insertion candidates, get the sequences, and carried out BLASTN alignments with the sp. 1 variants containing insertion candidates. If some variants

were found not to have the insertions, then the candidates were discarded. To filter deletion candidates, we similarly carried out BLASTN alignments with all the sp. 1 variants containing deletion candidates and selected those in which all sp. 1 variants had the deletions.

Step 5 is intended to identify and remove indels that correspond to introns that are shorter than 70 nt and is skipped in the permissive set. We first reexamined the BLASTN alignments made in step 1 and identified the residue numbers of the sp. 2 variants before and after each insertion. Then we determined if each indel candidate in sp. 1 corresponds to an intron in sp. 2. If the intron length so identified is less than 70 nt, then the indel candidate is discarded. The rest of indel candidates, including those that did not correspond to introns in sp. 2, were retained.

In step 6 only insertions or deletions in internal exons were selected.

### Generation mechanisms of fixed indels

An insertion is considered to be "possibly" generated by exonization if an intron whose length is more than or equal to two-thirds of the length of the insertion exists in sp. 2 at the corresponding location but is upgraded to "probably" generated by exonization if at least two-thirds of the residues were aligned by BLASTN (Fig 2A, C–E, G, and H). Likewise, a deletion is regarded as "possibly" generated by intronization if the corresponding segment in sp. 2 has an intron of at least two-thirds of the length of the deleted segment but is reclassified as "probably" generated by intronization if the BLASTN-aligned segment is more than or equal to two-thirds of the length of the deleted segment (Fig 2L, N–P, R, and S).

We identified the generation mechanisms of the rest of the fixed indels as follows. Homologous recombination is chosen as the generation mechanism of an insertion in sp. 1 based on the following conjecture (Fig S21A and B); sequence A with insertion generated by homologous recombination in sp. 1 is probably more like sequence A′ in sp. 1 that replaced the sequence than to the orthologous sequence B in sp. 2. Based on this idea, we set two criteria for an insertion to be a product of homologous recombination. First, there must be at least one sequence A′ of a different gene in sp. 1 aligned to the sp. 2 ortholog B, and second the fraction of the identity between the sequences of the insertion-containing exon and A′ is not significantly lower than that of the exon and B sequences (chi-square test). Similarly, a deletion in sp. 2 is considered to have been generated by homologous recombination if (1) sequence B in sp. 2 with an exon containing the deletion has at least one sequence B′ mapped to a different gene in sp. 2 that is aligned to the sp. 1 ortholog A and (2) the fraction of sequence identity between the exon and B′ is not significantly lower than that between the exon and A (chi-square test).

Tandem repeats were identified in coding sequences by Tandem Repeats Finder (Benson, 1999) with the mismatch and indel penalties and maximum period size set at 5, 3, and 2,000, respectively, whereas the other parameters were at default settings. We chose the loose parameter settings to identify as many tandem repeats as possible. If there are segments of tandem repeats in the fixed indels, the tandem repeats were regarded as responsible for generating the indels.

### Assignment of IDRs and statistical analyses

We ran IUPred3 (Erdős et al, 2021) and judged amino acids with a score more than or equal to 0.502 to be in IDRs. We carried out all the tests of statistical significance by $t$ test unless otherwise stated. For the data shown in Figs 5A, 8A, S7A, and S18, we first divided the frequency of each length bin by the average frequency of the species and arithmetically averaged the normalized values of the eight species.

## Data Availability

All data and code necessary to reproduce these analyses are available at Figshare; the lists of data and programs, the data of each selection step, the programs for the primates, rodents, *Oryzias* species, and *Drosophila* species, and the remaining four eukaryotes have been respectively deposited at https://figshare.com/articles/journal_contribution/Data_ProgramLists/29553317, https://figshare.com/articles/journal_contribution/DataFileModified/29553287, https://figshare.com/articles/journal_contribution/HumanChimpanzeeProgramsModified/29553299, https://figshare.com/articles/journal_contribution/MouseRatProgramsModified/29553305, https://figshare.com/articles/journal_contribution/MedakaProgramsModified/29553308, https://figshare.com/articles/journal_contribution/DrosophilaProgramsModified/29553311, and https://figshare.com/articles/journal_contribution/Other4SppProgramsModified/29553314.

## Supplementary Information

## Acknowledgements

This work was supported in part by Grant-in-Aid for Transformative Research Areas "Multifaceted proteins: Expanding and transformative protein world" from the Ministry of Education, Culture, Sports, Science and Technology in Japan, 20H05932.

### Author Contributions

K Homma: conceptualization, data curation, software, formal analysis, supervision, validation, investigation, methodology, project administration, and writing—original draft, review, and editing.
H Anbo: data curation, software, supervision, and writing—review and editing.
M Ota: supervision, funding acquisition, validation, and writing—review and editing.
S Fukuchi: supervision, funding acquisition, validation, and writing—review and editing.

## Conflict of Interest Statement

The authors declare that they have no conflict of interest.

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
