## [Reviewer comments · Life Science Alliance]

Longer internal exons tend to have more tandem repeats and more frequently experience indels

Keiichi Homma, Hiroto Anbo, Motonori Ota, and Satoshi Fukuchi

DOI: <https://doi.org/10.26508/lsa.202403148>

Corresponding author(s): Keiichi Homma, Maebashi Institute of Technology

Review Timeline:

Submission Date:	2024-11-26
Editorial Decision:	2025-03-14
Revision Received:	2025-07-14
Editorial Decision:	2025-08-26
Revision Received:	2025-09-07
Accepted:	2025-09-08

Scientific Editor: Tim Fessenden

Transaction Report:

March 14, 2025

Re: Life Science Alliance manuscript #LSA-2024-03148-T

Keiichi Homma
Maebashi Institute of Technology
Department of Life Science and Informatics
Maebashi 371-0816
Japan

Dear Dr. Homma,

Thank you for submitting your manuscript entitled "Longer internal exons tend to have more tandem repeats and more frequently experience insertions and deletions that are mostly in intrinsically disordered regions of the encoded proteins" to Life Science Alliance. The manuscript was assessed by expert reviewers, whose comments are appended to this letter. We invite you to submit a revised manuscript addressing the Reviewer comments.

Thank you for this interesting contribution to Life Science Alliance. We are looking forward to receiving your revised manuscript.

Sincerely,

B. MANUSCRIPT ORGANIZATION AND FORMATTING:

Reviewer #1 (Comments to the Authors (Required)):

Review of the Paper: "Longer internal exons tend to have more tandem repeats and more frequently experience insertions and deletions that are mostly in intrinsically disordered regions of the encoded proteins"

1. Summary of the Paper: The study investigates the relationship between long internal exons in eukaryotic genomes, the presence of tandem repeats, and the propensity for insertions and deletions (indels) in these regions. It demonstrates that indels, especially those encoding intrinsically disordered regions (IDRs), are more prevalent in long internal exons. The findings suggest that tandem repeat expansion contributed significantly to the evolutionary development of long exons and IDRs in eukaryotic proteins. The study proposes a model where tandem repeats and indels were central to shaping eukaryotic protein structures from the first eukaryotic common ancestor (FECA) to the last eukaryotic common ancestor (LECA).

2. Assessment of the Main Points: The data strongly support the correlation between tandem repeats and long internal exons. Statistical analyses and regression models demonstrate a positive association between exon length and tandem repeat prevalence. The study identifies tandem repeats as a primary contributor to the higher frequency of indels in long exons. No essential experiments are necessary, but additional cross-species validation could strengthen the claim. The evidence for indels in IDRs and frame preservation is robust, showing that most indels are non-frameshifting and tend to encode IDRs. This aligns with prior studies and adds novel insights into the evolutionary tolerability of indels in functional regions. More extensive sequence alignments from diverse taxa, including non-model organisms, could broaden the generalizability of the findings. The evolutionary model for the evolutionary role of tandem repeats in eukaryotic proteins proposed is consistent with observed patterns in the data, such as the preponderance of tandem repeats in eukaryotic proteins compared to prokaryotes. There could be more context about how the findings on tandem repeats and indels could influence our understanding of diseases or protein engineering. For example, mutations in IDRs are often linked to diseases, and understanding these evolutionary patterns could have biomedical implications. The authors could introduce unanswered questions in the field, such as the mechanisms regulating tandem repeat expansions or the factors influencing their tolerability in different organisms. Regarding methods, the authors could clarify the workflow by adding a flowchart or schematic diagram summarizing the workflow for identifying fixed indels and their classification. This will help readers visualize the step-by-step methodology and aid in potential reproducibility. They could also provide additional justification for thresholds used in identifying tandem repeats and indels (e.g., why 3 nt was chosen for alignment allowance). Some sections, particularly those describing methodologies, could benefit from simplification for accessibility to a broader audience. It would also be useful for the authors to discuss potential biases introduced by focusing on six species and how they might affect the generalizability of results. Consider including an analysis of other phylogenetic groups if feasible. Some sentences in the discussion and methods sections are overly complex. Simplifying these will make the paper more accessible to a broader audience. I would shorten the title of the paper as well.

3. In conclusion, this paper provides a meaningful contribution to understanding exon evolution and IDR formation in eukaryotic proteins. While the data are robust, expanding the scope to additional species and functional validations would solidify the conclusions and broaden their impact.

Reviewer #2 (Comments to the Authors (Required)):

In this manuscript, Homma et. al. performed a comprehensive characterization of the prevalence and mechanisms of formation of "fixed" indels by systematically comparing orthologous gene sequences for six species. Then, they identified compelling evidence that indels are enriched in long internal exons and are associated with the presence of IDR regions within those. Based on these analyses, the authors propose that indels, particularly tandem repeat expansions, have played a relevant role in the formation of long internal exons as seen in eukaryotic species from their common ancestors. Overall, although I find the scientific findings of relevance and the manuscript and presented analysis of quality, the manuscript would benefit from addressing the following queries:

1) Authors provide in Figure 3 six examples of fixed indels without providing a good reason or justification about why these examples were chosen to be described in greater detail. It would be very valuable if the authors provide additional evidence on the impact of these indels in protein structure and function (impact on protein folding, stability, alteration of protein domains? AlphaFold predictions can help here), serving to highlight their relevance.

2) The authors do not provide sufficient details regarding which genes are affected by the indels reported, aside from the six examples mentioned in the previous point. A thoughtful gene centric analysis of indel prevalence should be pursued. Are there indels recurrently affecting genes? If so, which genes are more frequently affected? are they enriched in a given molecular function or biotype (pathway or go term enrichment analysis could be pursued)? All these analyses may serve to provide additional hints regarding the functional consequences of the indels and associated IDR.

3) It is surprising to me that no indel associated with the insertion or deletion of a transposable element (TE) was found, given the prevalence of these repeats and their impact on the generation of genetic variation. Are not detected at all or are not detected as fixed indels? The vast majority of indels are of 1-6 base pairs, while transposable elements would be orders of magnitude bigger in size (e.g. 300 bp in the case of Alu or up to 6Kb for L1s). The authors should provide evidence to demonstrate that there are no technical limitations in the strategy used for indel detection that might account for this absence.

Additional issues:

4) In general, the text describing both the main and the supplementary figure captions is insufficient to ensure the message is self-contained. Please expand the text, providing further details, also the sample size must include both bar and line charts to enhance interpretability and ensure transparency. I had to go in multiple instances to the supplementary tables to know what is the sample size of a given distribution to understand that it was not significant due to low N.

5) I believe that the reference to Fig. 5 refers to Fig. 6 at line number 13 of page 13.

6) Rephrase the following sentence as unclear: "Indels have amino acid compositions similar to those of IDRs in that order-promoting residues are depleted". Line number 4th of page 13.

Reviewer #1 (Comments to the Authors (Required)):

First of all, we deeply regret that we found and corrected two bugs in the analyses presented in the last manuscript. One bug is concerned with the length of internal exons that have introns; although the length the exon is supposed to be the length BEFORE the insertion, the length after the insertion was used due to a bug in our programs. The debugging resulted in shorter lengths corresponding to insertions. Particularly, high peaks appeared in the 1-30 nt range in many species because a small number of indels (numerator) gives rise to aberrantly high frequencies due to a very small number of internal exons in this length range (denominator). The second bug is that the length of transcript ID we used before was too short for many rat and mouse transcripts, resulting in the failure to identify many indels. The correction increased the indel frequency ~4 fold in rat and mouse exons but did not significantly affect the overall trends.

To cope with the aberrantly high indel frequencies in the 1-30 nt range, we switched to 60 from 30 nt exon length bins for our analyses. We essentially got the same results as in the original version, except for the insertion frequencies in the two fish species added to our analyses (see below). To check if the choice of length bins does not affect results, we also calculated indel frequencies in three 180 nt internal exon length ranges (S: 1-180 nt, M: 181-360 nt, and L: 361-540 nt) and compared them. As the generally increasing trends in indel frequencies with internal exons are supported in the aggregated results, we consider it unlikely that our conclusions are affected by the choice of the 60 nt length bins.

Following reviewer #1's suggestion, we identified and analyzed indels in two fish species. Although the deletion frequencies showed the same increasing trend with internal exon length as the other six eukaryotes, the insertion frequencies remain essentially unchanged regardless of internal exon length, unlike the rest. Since insertion frequencies in longer exons are higher in six out of eight species, however, we did not alter the general observation but moderated expressions. Future research will reveal whether these fish species are exceptions as we interpret.

Since the number of deleted residues was much higher than that of inserted residues in the fish species, the average deletion-to-insertion ratio became much larger (4.21) than before. If we were to multiply deletion frequencies by this factor in Fig 5A as we did in the original version, the trend in insertion would be hard to see. We therefore chose not to multiply deletion frequencies in Fig 5A.

Review of the Paper: "Longer internal exons tend to have more tandem repeats and more

frequently experience insertions and deletions that are mostly in intrinsically disordered regions of the encoded proteins"

1. Summary of the Paper: The study investigates the relationship between long internal exons in eukaryotic genomes, the presence of tandem repeats, and the propensity for insertions and deletions (indels) in these regions. It demonstrates that indels, especially those encoding intrinsically disordered regions (IDRs), are more prevalent in long internal exons. The findings suggest that tandem repeat expansion contributed significantly to the evolutionary development of long exons and IDRs in eukaryotic proteins. The study proposes a model where tandem repeats and indels were central to shaping eukaryotic protein structures from the first eukaryotic common ancestor (FECA) to the last eukaryotic common ancestor (LECA).

The summary made by the reviewer elegantly summarizes the essence of the manuscript.

2. Assessment of the Main Points: The data strongly support the correlation between tandem repeats and long internal exons. Statistical analyses and regression models demonstrate a positive association between exon length and tandem repeat prevalence. The study identifies tandem repeats as a primary contributor to the higher frequency of indels in long exons. No essential experiments are necessary, but additional cross-species validation could strengthen the claim. The evidence for indels in IDRs and frame preservation is robust, showing that most indels are non-frameshifting and tend to encode IDRs. This aligns with prior studies and adds novel insights into the evolutionary tolerability of indels in functional regions. More extensive sequence alignments from diverse taxa, including non-model organisms, could broaden the generalizability of the findings.

Thank you for the comment. We note that the use of non-model organisms to identify fixed indels may misidentify many alternatively splicing variants as indels. This is because the uniformity requirements to identify fixed indels do not work if the number of variants is small. We need to have three closely related species with nearly all of their variants sequenced to accurately select fixed indels. With this potential danger in mind, we identified and carried out analyses of indels in two fish species. As the insertions in internal exons in the two species are apparently not correlated with their length, we toned down our statement that insertions occur more frequently in longer exons.

2 (ctd)The evolutionary model for the evolutionary role of tandem repeats in eukaryotic proteins proposed is consistent with observed patterns in the data, such as the preponderance of tandem repeats in eukaryotic proteins compared to prokaryotes. There could be more context about how

the findings on tandem repeats and indels could influence our understanding of diseases or protein engineering. For example, mutations in IDRs are often linked to diseases, and understanding these evolutionary patterns could have biomedical implications.

We appreciate the suggestions We carried out gene-ontology (GO) analyses of human and mouse indels as these are the only organisms out of the eight whose genes have been adequately annotated in SwissProt. In addition, we investigated if any of the human insertions and chimpanzee deletions are located in disease-related sites of the proteins as annotated in SwissProt. Remarkably some indels were found to be in disease-related sites. The results are presented in Discussion.

2 (ctd) The authors could introduce unanswered questions in the field, such as the mechanisms regulating tandem repeat expansions or the factors influencing their tolerability in different organisms.

Thank you for the comments. As our results indicate that slipped strand mispairing rather than homologous recombination accounts for most of repeat expansions, we added a description to the Discussion section (first paragraph). The tolerability of tandem repeats in prokaryotes is likely to be lower than that of eukaryotes because the former generally contain a lower fraction of tandem repeats as well as IDRs than the latter. We have inserted the argument.

2 (ctd) Regarding methods, the authors could clarify the workflow by adding a flowchart or schematic diagram summarizing the workflow for identifying fixed indels and their classification. This will help readers visualize the step-by-step methodology and aid in potential reproducibility.

The figure presented as Supplementary Figure S1 is meant as a flowchart. We added detailed methods to Material and Methods to clarify our methods.

2 (ctd) They could also provide additional justification for thresholds used in identifying tandem repeats and indels (e.g., why 3 nt was chosen for alignment allowance).

Thank you for the comments. We added explanations of why we chose the parameters of Tandem Repeats Finder and of why we chose 3 nt for alignment allowance.

2 (ctd) Some sections, particularly those describing methodologies, could benefit from

simplification for accessibility to a broader audience.

We modified the Materials and Method section to improve readability.

2 (ctd) It would also be useful for the authors to discuss potential biases introduced by focusing on six species and how they might affect the generalizability of results. Consider including an analysis of other phylogenetic groups if feasible.

Following the suggestion, we identified and analyzed indels in two fish species. The results were added to the manuscript.

2 (ctd) Some sentences in the discussion and methods sections are overly complex. Simplifying these will make the paper more accessible to a broader audience. I would shorten the title of the paper as well.

We modified some sentences to increase readability. We acknowledge that the title in the original manuscript was too long. The new title is “Longer internal exons tend to have more tandem repeats and more frequently experience indels”.

3. In conclusion, this paper provides a meaningful contribution to understanding exon evolution and IDR formation in eukaryotic proteins. While the data are robust, expanding the scope to additional species and functional validations would solidify the conclusions and broaden their impact.

We greatly appreciate the encouraging comments.

Reviewer #2 (Comments to the Authors (Required)):

First of all, we deeply regret that we found and corrected two bugs in the analyses presented in the last manuscript. One bug is concerned with the length of internal exons that have introns; although the length the exon is supposed to be the length BEFORE the insertion, the length after the insertion was used due to a bug in our programs. The debugging resulted in shorter lengths corresponding to insertions. Particularly, high peaks appeared in the 1-30 nt range in many species because a small number of indels (numerator) gives rise to aberrantly high frequencies

due to a very small number of internal exons in this length range (denominator). The second bug is that the length of transcript ID we used before was too short for many rat and mouse transcripts, resulting in the failure to identify many indels. The correction increased the indel frequency ~4 fold in rat and mouse exons but did not significantly affect the overall trends.

To cope with the aberrantly high indel frequencies in the 1-30 nt range, we switched to 60 from 30 nt exon length bins for our analyses. We essentially got the same results as in the original version, except for the insertion frequencies in the two fish species added to our analyses (see below). To check if the choice of length bins does not affect results, we also calculated indel frequencies in three 180 nt internal exon length ranges (S: 1-180 nt, M: 181-360 nt, and L: 361-540 nt) and compared them. As the generally increasing trends in indel frequencies with internal exons are supported in the aggregated results, we consider it unlikely that our conclusions are affected by the choice of the 60 nt length bins.

Following reviewer #1's suggestion, we identified and analyzed indels in two fish species. Although the deletion frequencies showed the same increasing trend with internal exon length as the other six eukaryotes, the insertion frequencies remain essentially unchanged regardless of internal exon length, unlike the rest. Since insertion frequencies in longer exons are higher in six out of eight species, however, we did not alter the general observation but moderated expressions. Future research will reveal whether these fish species are exceptions as we interpret.

Since the number of deleted residues was much higher than that of inserted residues in the fish species, the average deletion-to-insertion ratio became much larger (4.21) than before. If we were to multiply deletion frequencies by this factor in Fig 5A as we did in the original version, the trend in insertion would be hard to see. We therefore chose not to multiply deletion frequencies in Fig 5A.

In this manuscript, Homma et. al. performed a comprehensive characterization of the prevalence and mechanisms of formation of "fixed" indels by systematically comparing orthologous gene sequences for six species. Then, they identified compelling evidence that indels are enriched in long internal exons and are associated with the presence of IDR regions within those. Based on these analyses, the authors propose that indels, particularly tandem repeat expansions, have played a relevant role in the formation of long internal exons as seen in eukaryotic species from their common ancestors. Overall, although I find the scientific findings of relevance and the manuscript and presented analysis of quality, the manuscript would benefit from addressing the

following queries:

1) Authors provide in Figure 3 six examples of fixed indels without providing a good reason or justification about why these examples were chosen to be described in greater detail. It would be very valuable if the authors provide additional evidence on the impact of these indels in protein structure and function (impact on protein folding, stability, alteration of protein domains? Alphafold predictions can help here), serving to highlight their relevance.

The six examples in Fig. 3 were chosen not because they are structurally interesting, but because they enable the reader to easily visualize what kinds of indels were selected. We have modified the text to make this clear.

2) The authors do not provide sufficient details regarding which genes are affected by the indels reported, aside from the six examples mentioned in the previous point. A thoughtful gene centric analysis of indel prevalence should be pursued. Are there indels recurrently affecting genes? If so, which genes are more frequently affected? are they enriched in a given molecular function or biotype (pathway or go term enrichment analysis could be pursued)? All these analyses may serve to provide additional hints regarding the functional consequences of the indels and associated IDR.

There are genes with multiple indels as are listed in the data provided online. Following the good suggestion, we carried out gene ontology (GO) analyses of genes with indels in humans and mice, as these are the only ones out of the eight species to which adequate SwissProt annotations are available. The results appear in the Results and Table S6.

3) It is surprising to me that no indel associated with the insertion or deletion of a transposable element (TE) was found, given the prevalence of these repeats and their impact on the generation of genetic variation. Are not detected at all or are not detected as fixed indels? The vast majority of indels are of 1-6 base pairs, while transposable elements would be orders of magnitude bigger in size (e.g. 300 bp in the case of Alu or up to 6Kb for L1s). The authors should provide evidence to demonstrate that there are no technical limitations in the strategy used for indel detection that might account for this absence.

There were several genes with indels that were found to contain TEs, but no indels overlapped with TEs. As we used the entire gene sequences in running BLASTN alignments, TEs in indels must have been detected with the lax criterion, if they do exist without many mutations. We added a possible reason for the absence of TEs in fixed indels; TEs in exons mostly occur in

alternatively spliced exons and these may have been removed in our methodology to identify fixed indels commonly observed in all variants.

Additional issues:

4) In general, the text describing both the main and the supplementary figure captions is insufficient to ensure the message is self-contained. Please expand the text, providing further details, also the sample size must include both bar and line charts to enhance interpretability and ensure transparency. I had to go in multiple instances to the supplementary tables to know what is the sample size of a given distribution to understand that it was not significant due to low N.

We have expanded the text and figure captions to make it easier to understand and added a bar figure to show numbers.

5) I believe that the reference to Fig. 5 refers to Fig. 6 at line number 13 of page 13.

We are referring to the generally higher prevalence of indels in longer internal exons here and Fig. 5 is the one that graphically shows the trend.

6) Rephrase the following sentence as unclear: "Indels have amino acid compositions similar to those of IDRs in that order-promoting residues are depleted". Line number 4th of page 13.

We appreciate the comment and acknowledge that the original sentence was hard to understand. The rephrased sentence is "Indels have amino acid compositions similar to those of IDRs; in indels order-promoting residues are depleted, while disorder-promoting residues appear more frequently than the average."

August 26, 2025

RE: Life Science Alliance Manuscript #LSA-2024-03148-TR

Dr. Keiichi Homma
Maebashi Institute of Technology
Department of Life Science and Informatics
460-1 Kamisadori-machi
Maebashi 371-0816
Japan

Dear Dr. Homma,

Thank you for submitting your revised manuscript entitled "Longer internal exons tend to have more tandem repeats and more frequently experience indels". As you will see, reviewers are overall satisfied with no further major concerns. We concur with the suggestion by Reviewer 2 to include sample sizes in each figure caption (or on the figure panel itself). Please note that adding a text label to indicate each figure, per the suggestion from Reviewer 2, is not recommended for publication. Finally, we invite you to incorporate the suggestions from this reviewer to extend the discussion of these findings (points 2.1 - 2.3) if you wish. We would be happy to publish your paper in Life Science Alliance pending these changes as well as final revisions necessary to meet our formatting guidelines.

- Please add the X and Bluesky handles of your host institute/organization as well as your own or/and one of the authors in our system.
- Please provide a manuscript file/table files without tracked-changes/highlights.
- Please remove labels 'Figure 1 goes in here' etc. from the main manuscript text
- Please add your main, supplementary figure, and table legends to the main manuscript text after the references section.
- Please ensure that all panels present in each figure are mentioned in the figure caption: labels A-V are missing in the caption of Figure 2; labels H-I are missing in the caption of Figure 5; labels I-J are missing in the caption of Figure S3; labels A-I are missing in the caption of Figure S7; labels A-H Figure S2; labels A-B Figure S4; labels A-I Figure S6; A-I Figure S8; A-I Figure S9; A-B Figures S10 and S11; A-H Figure S14; A-I figure S15; A-I Figure S17; A-B Figure S19 and S21.
- Please adjust the callouts in the manuscript accordingly as per the point above. Each figure panel must be cited in manuscript text: e.g. Figure 5A-I missing, Figure S2A-H etc.

A. FINAL FILES:

-- Summary blurb (enter in submission system): A short text summarizing in a single sentence the study (max. 200 characters including spaces). This text is used in conjunction with the titles of papers, hence should be informative and complementary to the title. It should describe the context and significance of the findings for a general readership; it should be written in the

present tense and refer to the work in the third person. Author names should not be mentioned.

B. MANUSCRIPT ORGANIZATION AND FORMATTING:

Thank you for your attention to these final processing requirements. Please revise and format the manuscript and upload materials as soon as you are able.

Sincerely,

Reviewer #1 (Comments to the Authors (Required)):

The authors have made a commendable effort in addressing the reviewers' comments with clarity and transparency. The acknowledgment and correction of the two bugs in the analysis are appreciated, particularly the careful explanation of how these corrections influenced the results without undermining the overall conclusions. The addition of two fish species and the revised treatment of exon length bins effectively strengthen the analysis and account for potential artifacts. The inclusion of gene ontology analysis, especially for indels in disease-related regions, provides valuable biological context and enhances the relevance of the findings. Additionally, the expanded discussion on mechanisms of repeat expansion and the improved methodological descriptions-including a more accessible Materials and Methods section and clarified figure captions-significantly improve the paper's rigor and readability. The authors' willingness to moderate interpretations where appropriate (e.g., in fish insertion frequencies) reflects thoughtful improvements to the manuscript. Overall, the revisions substantially improve the manuscript and address the reviewers' biggest concerns.

Reviewer #2 (Comments to the Authors (Required)):

SECOND REVIEW FOR "Longer internal exons tend to have more tandem repeats and more frequently experience indels" Homma et al. have submitted a substantially revised manuscript that effectively addresses the majority of the reviewers' comments. Notably, the authors have strengthened the study by including two additional fish species, conducting gene ontology analysis of genes with indels, and examining the overlap with disease-associated protein residues. They have also improved the overall readability of the manuscript, clarified the methodology, and provided stronger justifications for their methodological choices. Additionally, the authors have transparently reported an issue in their original analysis, which affected the initially submitted version. This issue has since been corrected, and the authors have demonstrated that it does not compromise their original findings or conclusions.

While these revisions are appreciated and have improved the manuscript, I believe that further refinement is needed in two key areas: (1) enhancing the transparency in the presentation of data and results across both the main and supplementary figures,

and (2) more clearly articulating and discussing the potential biological implications of the association between IDR and tandem repeats in internal long exons. These were concerns raised in the initial review, and although the authors have made efforts to address them, I find the current revisions insufficient in fully resolving these points.

(1) Multiple figures use bar charts with percentages on the Y-axis (e.g., Figures 4 and 6; Supplementary Figures 3, 5, 6, and 10-16). While this type of visualization is useful for highlighting relative differences, it is highly sensitive to variations in sample size. To improve transparency, sample sizes should be indicated above each bar, as done in Supplementary Figure 2. Where this is not feasible, the sample size should be clearly stated in the figure caption. Additionally, the manuscript currently presents main and supplementary figures without explicitly labeling which is which. Although the figures appear in order, this lack of clear labeling makes it difficult to reference them accurately, so I may have made errors in the figure numbers cited above.

(2.1) Based on the online data, it is not clear which orthologous genes are recurrently affected by indels across different taxa. This information is important, as it could highlight genes under relaxed selective constraint or those with recurrently evolving intrinsically disordered regions (IDRs) driven by indels. Such findings would be valuable for readers interested in the functional or evolutionary implications of the results. A summary of these recurrently affected orthologs should be included in the main text to make these insights more accessible.

(2.2) The authors should further interpret and contextualize the interesting findings from the GO enrichment analysis, which is now included in the main text. For example, recent studies have demonstrated the functional relevance of IDRs in acetyltransferases and histone modification pathways (e.g., Matsuoka et al., 2024; PMID: 38777743). Drawing cautious connections between these findings and the enriched GO terms could help strengthen the biological relevance of the observed associations, without overstating their significance.

(2.3) A similar suggestion applies to the reported indels overlapping disease-associated protein residues—specifically, the 6 insertions and 12 deletions identified. A more in-depth analysis focusing on a few representative or particularly interesting cases would add considerable value. Such an exploration could help illustrate potential mechanistic consequences and better demonstrate the functional impact of these events.

Reviewer #1 (Comments to the Authors (Required)):

(Our comments) Since Figure S19 in the previous version was erroneously based on data of six species without the two fish species, we redrew the figure using data of the eight species. Our apologies. Fortunately, the correction did not affect the conclusion.

The authors have made a commendable effort in addressing the reviewers' comments with clarity and transparency. The acknowledgment and correction of the two bugs in the analysis are appreciated, particularly the careful explanation of how these corrections influenced the results without undermining the overall conclusions. The addition of two fish species and the revised treatment of exon length bins effectively strengthen the analysis and account for potential artifacts. The inclusion of gene ontology analysis, especially for indels in disease-related regions, provides valuable biological context and enhances the relevance of the findings. Additionally, the expanded discussion on mechanisms of repeat expansion and the improved methodological descriptions-including a more accessible Materials and Methods section and clarified figure captions-significantly improve the paper's rigor and readability. The authors' willingness to moderate interpretations where appropriate (e.g., in fish insertion frequencies) reflects thoughtful improvements to the manuscript. Overall, the revisions substantially improve the manuscript and address the reviewers' biggest concerns.

(Our comments) Thank you very much for the kind words. We were fortunate to have constructive criticisms of the first manuscript that led us to improve the paper.

Reviewer #2 (Comments to the Authors (Required)):

SECOND REVIEW FOR "Longer internal exons tend to have more tandem repeats and more frequently experience indels"

(Our comments) Since Figure S19 in the previous version was erroneously based on data of six species without the two fish species, we redrew the figure using data of the eight species. Our apologies. Fortunately, the correction did not affect the conclusion.

Homma et al. have submitted a substantially revised manuscript that effectively addresses the majority of the reviewers' comments. Notably, the authors have strengthened the study by including two additional fish species, conducting gene ontology analysis of genes with indels, and examining the overlap with disease-associated protein residues. They have also improved the overall readability of the manuscript, clarified the methodology, and provided stronger justifications for their methodological choices. Additionally, the authors have transparently reported an issue in their original analysis, which affected the initially submitted version. This issue has since been corrected, and the authors have demonstrated that it does not compromise

their original findings or conclusions.

(Our comments) Thank you very much for the warm words. We were fortunate to have constructive criticisms of the first manuscript that led us to substantially improve the paper.

While these revisions are appreciated and have improved the manuscript, I believe that further refinement is needed in two key areas: (1) enhancing the transparency in the presentation of data and results across both the main and supplementary figures, and (2) more clearly articulating and discussing the potential biological implications of the association between IDR and tandem repeats in internal long exons. These were concerns raised in the initial review, and although the authors have made efforts to address them, I find the current revisions insufficient in fully resolving these points.

(1) Multiple figures use bar charts with percentages on the Y-axis (e.g., Figures 4 and 6; Supplementary Figures 3, 5, 6, and 10-16). While this type of visualization is useful for highlighting relative differences, it is highly sensitive to variations in sample size. To improve transparency, sample sizes should be indicated above each bar; as done in Supplementary Figure 2. Where this is not feasible, the sample size should be clearly stated in the figure caption.

(Our comments) We acknowledge that it is important to display sample numbers in the figures and have added the information to all the figures.

Additionally, the manuscript currently presents main and supplementary figures without explicitly labeling which is which. Although the figures appear in order, this lack of clear labeling makes it difficult to reference them accurately, so I may have made errors in the figure numbers cited above.

(Our comments) We have modified the manuscript to specifically state panel(s) of each figure the text refers to. We also expanded the figure legends to clarify what each graph represents. Hopefully these changes have enhanced the clarity of our manuscript.

(2.1) Based on the online data, it is not clear which orthologous genes are recurrently affected by indels across different taxa. This information is important, as it could highlight genes under relaxed selective constraint or those with recurrently evolving intrinsically disordered regions (IDRs) driven by indels. Such findings would be valuable for readers interested in the functional or evolutionary implications of the results. A summary of these recurrently affected orthologs should be included in the main text to make these insights more accessible.

(Our comments) Our indel lists consist of probable cases of indels but do not include all indel cases in the species. The identification rates of indels considerably vary mainly because the

evolutionary distances among the trio of species differ among the eight species. For one thing, the number of indels identified in *Homo sapiens* is much smaller than that in *Rattus norvegicus*. We thus consider the current data insufficient for identifying orthologous genes commonly affected by indels. We have made the tables of indels with gene names publicly available so that interested researchers can conduct analysis.

(2.2) The authors should further interpret and contextualize the interesting findings from the GO enrichment analysis, which is now included in the main text. For example, recent studies have demonstrated the functional relevance of IDRs in acetyltransferases and histone modification pathways (e.g., Matsuoka et al., 2024; PMID: 38777743). Drawing cautious connections between these findings and the enriched GO terms could help strengthen the biological relevance of the observed associations, without overstating their significance.

(Our comments) We appreciate the comments as we agree that the GO enrichment analysis presented in the previous manuscript was inadequate. Our interest is not so much specific functions affected by indels as the general evolutionary roles played by intrinsically disordered regions (IDRs). We thus took a close look at the results of the GO enrichment analysis and found that functions related to the nucleus constitute half of them. Since IDRs are prevalent in proteins localized to the nucleus, we added our interpretations of the results in terms of IDRs. As we have made our data freely available, other researchers can analyze indels identified by our investigation according to their interest.

(2.3) A similar suggestion applies to the reported indels overlapping disease-associated protein residues-specifically, the 6 insertions and 12 deletions identified. A more in-depth analysis focusing on a few representative or particularly interesting cases would add considerable value. Such an exploration could help illustrate potential mechanistic consequences and better demonstrate the functional impact of these events.

(Our comments) We also acknowledge the inadequacy of the original presentation of the indels that coincide with disease-related sites. As our main interests lie not in specific human genes affected, but in general roles IDRs play in evolution, we examined which sites are within IDRs. We found that three quarters of disease-related indels sites encode IDRs and added description of this to the manuscript. As described in Discussion, this finding agrees with previous reports that IDRs are frequently involved in human diseases. We thank the reviewer for giving us an opportunity to make a better thought-out interpretation of our analysis.

September 8, 2025

RE: Life Science Alliance Manuscript #LSA-2024-03148-TRR

Dr. Keiichi Homma
Maebashi Institute of Technology
Department of Life Science and Informatics
460-1 Kamisadori-machi
Maebashi 371-0816
Japan

Dear Dr. Homma,

Thank you for submitting your Research Article entitled "Longer internal exons tend to have more tandem repeats and more frequently experience indels". It is a pleasure to let you know that your manuscript is now accepted for publication in Life Science Alliance. Congratulations on this interesting work.

DISTRIBUTION OF MATERIALS:

Again, congratulations on a very nice paper. I hope you found the review process to be constructive and are pleased with how the manuscript was handled editorially. We look forward to future exciting submissions from your lab.

Sincerely,
